# Semi-supervised Node Importance Estimation with Informative Distribution Modeling for Uncertainty Regularization

## ABSTRACT

*Graph node importance estimation*, a classical problem in network analysis, underpins various web applications. To improve estimation accuracy, previous methods either exploit intrinsic topological characteristics, e.g., graph centrality, or leverage additional information, e.g., data heterogeneity, for node feature enhancement. However, these methods follow the *supervised learning* setting, overlooking the fact that ground-truth node-importance data are usually partially labeled in practice. In this work, we propose the first semi-supervised node importance estimation framework, i.e., EASING, to improve learning quality for unlabeled data in heterogeneous graphs. Different from previous approaches, EASING explicitly captures *uncertainty* to reflect the confidence of model predictions. To jointly estimate the importance values and uncertainties, EASING incorporates DJE, a deep encoder-decoder neural architecture. DJE introduces *distribution modeling* for graph nodes, where the distribution representations are decoded to derive both importance and uncertainty estimates, after encoding the rich heterogeneous graph information. Additionally, DJE facilitates effective pseudo-label generation for the unlabeled data to enrich the training samples. Then based on both labeled and pseudo-labeled data, EASING develops effective semi-supervised heteroscedastic learning with the varying node uncertainty regularization. Extensive experiments on three real-world datasets highlight the superior performance of EASING compared to competing methods and demonstrate the effectiveness of each individual module. Codes are available via https://anonymous.4open.science/r/EASING-2F70/.

## KEYWORDS

Node Importance Estimation, Semi-supervised Learning, Heterogeneous Graph, Network Analysis, Uncertainty Regularization

**ACM Reference Format:**
Anonymous authors. 2018. Semi-supervised Node Importance Estimation with Informative Distribution Modeling for Uncertainty Regularization. In *Proceedings of Make sure to enter the correct conference title from your rights confirmation emai (Conference acronym 'XX).* ACM, New York, NY, USA

## 1 INTRODUCTION

*Node importance estimation* is a fundamental problem in network science, as it reveals the significance of individual nodes by evaluating both their intrinsic properties and relationships with others. It forms

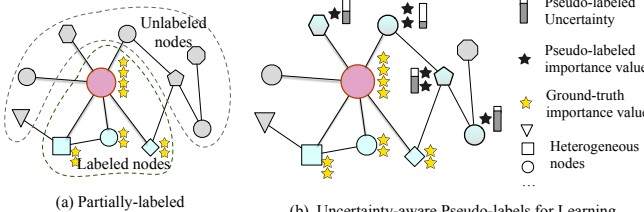

**Figure 1: With existing partially labeled data, i.e., indicated by golden stars, EASING further constructs pseudo-labeled importance values (in black) and uncertainties (in gray) for semi-supervised learning.**

the basis for various Web applications, such as key-opinion-leader discovery, social network analysis, query disambiguation, medical knowledge analysis, and network resource allocation [26, 33]. Early algorithm-based approaches focus on graph topology analysis, such as node degree centrality and Pagerank methodologies [23, 24].

The learning-based methods leverage deep learning techniques to capture the rich and diverse graph information, e.g., in heterogeneous graphs, for node importance learning and quantitation. To learn the high-order topology knowledge, these methods usually employ graph-based neural networks to enrich and vectorize the latent and heterogeneous features of graph nodes. For instance, the pioneer work GENI [25] applies attention mechanism to aggregate the structure information for node importance estimation. Subsequent work [17, 26] utilizes a variety of information enrichment for model performance improvement. However, they focus on solving the *importance-based ranking problem*, instead of quantifying exact importance values. Recent research [5, 16] recognizes the importance value heterogeneity, where different node types can represent varied semantics and value ranges. Consequently, they propose dedicated designs to jointly consider local-global structural and textual information, to achieve competitive performance in *node importance value estimation*. All these works operate within the *supervised learning setting* to rely solely on labeled data, i.e., graph nodes with ground-truth importance values. However, due to the highly graph heterogeneity, the labeled graph nodes typically tend to be scarce, as accurate annotation can be costly in practice. For instance, one of the widely-studied datasets, namely TMDB5K, is annotated with only around 4% of all graph nodes [17, 25, 26]. Learning from partially labeled data could be inadequate for effective model training and lead to sub-optimal performance accordingly.

One potential solution to this issue is to investigate the problem in the *semi-supervised learning (SSL) setting*, which leverages additional supervisory signals from unlabeled graph nodes for optimization [31]. While SSL holds great promise, it is non-trivial to apply conventional SSL methods, e.g., via pseudo-label supervision, in the context of node importance estimation. The challenges are mainly twofold. (1) Compared to ground-truth node importance labels, learning from unlabeled graph nodes may inevitably introduce

noise that biases the model optimization. (2) The studied problem is essentially formulated as regression, indicating that the expected pseudo-labels are *continuous numerical values*, rather than the *binary 1/0 labels* in SSL classification problems that can be smoothly obtained via thresholding functions [29, 32].

To tackle these challenges, we propose the framework namely *s*Emi-supervised node importAnce eStimatIon with uNcertainty reGularization (EASING). (1) EASING explicitly considers the concept of "uncertainty" to indicate the confidence in model predictions. In this work, the uncertainty is solely determined by the graph node information, without considering external factors. We then leverage the estimated uncertainty to regularize the semi-supervised learning process. In contrast to the conventional homoscedastic setting, where each data sample equally contributes to loss accumulation, our semi-supervised *heteroscedastic* learning paradigm assigns varying weights to node samples based on their uncertainty. (2) To jointly estimate the importance value and its associated uncertainty, we propose an effective deep neural network namely `Distribution-based Joint Estimator (DJE)`. DJE is based on the assumption that each graph node follows a *stochastic distributions*, which naturally incorporates uncertainty while providing flexibility in aggregating graph information. DJE adopts an encoder-decoder structure, where the encoder captures rich heterogeneous graph information to represent node distributions, while the decoder then utilizes the distribution mean and covariance to derive the target importance and uncertainty. (3) Given that the labels in our problem are in the numerical format, we generate pseudo-labeled pairs of node importance and uncertainty for the unlabeled data. This is achieved by directly ensembling DJE's predictions with variational inference. The quality of such generated pseudo-labels is further analyzed. As shown in Figure 1, by utilizing both the ground-truth labeled data and constructed pseudo-labeled data, EASING facilitates more effective model training.

We conduct extensive experiments on three real-world datasets that have been widely evaluated in prior work [5, 16, 17, 25, 26]. The empirical analyses demonstrate not only the performance superiority of our EASING compared to competing methods, but also the effectiveness of all the modules contained therein. To summarize, our primary contributions are threefold as follows:

- We propose EASING with uncertainty regularization, which, to the best of our knowledge, is the first semi-supervised work for node importance estimation in heterogeneous graphs.
- We propose DJE, an encoder-decoder neural architecture that (1) delivers informative distribution modeling for joint estimation of both importance values and uncertainties, and (2) enables effective pseudo-label generation.
- Extensive experiments on three widely-studied real-world datasets demonstrate the effectiveness of both our EASING framework as well as its constituent modules.

## 2 RELATED WORK

### 2.1 Graph Node Importance Estimation

Early traditional solutions mainly work on graph topology analysis, such as centrality measures [22, 23, 28], PageRank [24], Degree [23], Closeness [28], Eigenvector [22], and Harmonic [21]. These methods are all proposed for homogeneous graphs. With

the development of deep learning techniques, a series of learning-based methods recently have been proposed for heterogeneous graphs with different types of information included. For example, GENI [25] and MULTIIMPORT [26] infer the node importance by using multi-relational graph structure information. RGTN [17] considers both structural and semantic information to estimate node importance. However, these works concentrate on ranking node importance without estimating the specific values of nodes. Recently, HIVEN [16] firstly considers the value heterogeneity in heterogeneous information networks with local and global modules. SKES [5] exploits deep graph structure information and optimal transport theory to estimate the value of the node importance. All these proposed methods are within the supervised learning setting, solely relying on the labeled data for model training.

### 2.2 Semi-supervised Learning for Graph Data

Semi-supervised Learning (SSL) aims to solve the problems that a few samples are labeled but most of them are not, due to the difficult, expensive, or time-consuming labeling process [30, 31]. SSL methods have been applied to many domains, including CV, NLP, etc. Recently, a few works is starting the study of SSL methodologies, particularly for graph data to exploit the rich topological information. Specifically, GRAND [13] studies for homogeneous graphs, augmenting graph data by a random propagation strategy to improve the performance in the SSL setting. MoDis [27] constructs pseudo-labels by summarizing the disagreements of the model's predictions. For heterogeneous graphs, Meta-PN [12] adopts the strategy of pseudo-label construction by applying a meta-learning label propagation approach to learn high-quality pseudo-labels. HG-MDA [6] combines multi-level data augmentation and meta-relation-based attention to capture information from different types of nodes and edges. Nevertheless, all these works aim to identify the type of nodes for classification, instead of the regression problems, e.g., estimating the node importance value. As pointed out by [29, 32], regression problems differ fundamentally from classification problems as they produce predictions in the form of real numbers rather than class probabilities. Current semi-supervised classification techniques are unsuitable for semi-supervised regression since they depend on class probabilities and thresholding functions to create pseudo-labels. Consequently, a straightforward conversion may be non-trivial.

## 3 PRELIMINARY: SSL FOR NODE IMPORTANCE ESTIMATION

A common framework in semi-supervised learning is to construct the objectives for both label and unlabeled data [30, 31]. For all labeled data $\mathcal{D} = \{(x, s_x)\}$ and unlabeled nodes $\mathcal{D}' = \{x'\}$, we use $s_x$ to denote the ground-truth label of node $x$, which in our problem is the node importance value. Their corresponding loss terms, i.e., $\mathcal{L}^{lb}$ and $\mathcal{L}^{unlb}$, are jointly optimized with the weight $\lambda$ as:

$$\mathcal{L} = \mathcal{L}^{lb} + \lambda \mathcal{L}^{unlb}. \tag{1}$$

To customize the framework for the *node importance estimation problem*, which is essentially a regression problem, one straightforward solution is to incorporate the regression loss such as mean

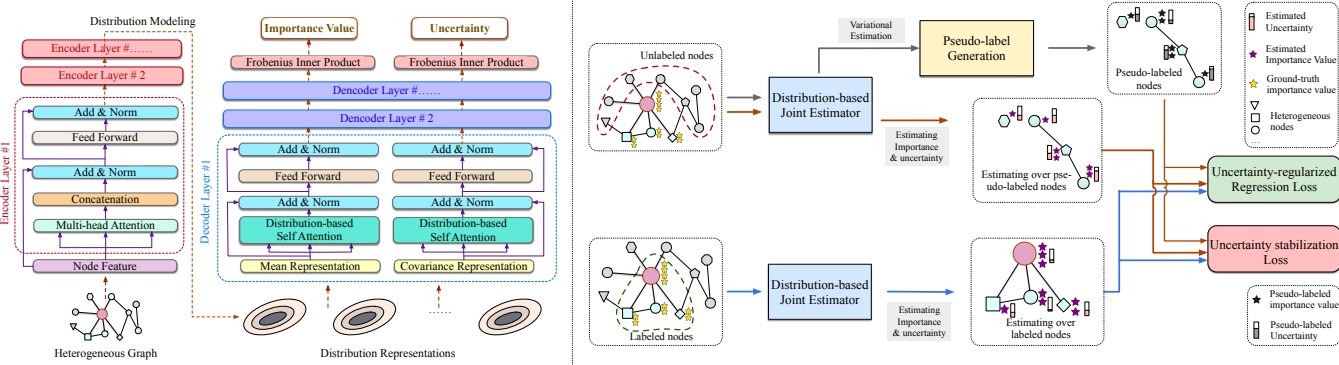

(A) Distribution-based Joint Estimator (DJE) with an encoder-decoder architecture.    (B) Semi-supervised Heteroscedastic Learning with Uncertainty Regularization.

Figure 2: An illustration of our EASING framework with (A) DJE structure and (B) uncertainty-regularized learning flow.

squared error (MSE) for the supervised learning of labeled data as:

$$\mathcal{L} = \frac{1}{|\mathcal{D}|} \sum_{x \in \mathcal{D}} (s_x - \widehat{s}_x)^2 + \lambda \mathcal{L}^{unlb}, \tag{2}$$

Where $\widehat{s}_x$ denotes the estimated node importance value. In SSL, $\mathcal{L}^{unlb}$ is used to extract the weak supervision signal from unlabeled data for model optimization. Among various SSL methods, we focus on generating pseudo-labels for optimizing $\mathcal{L}^{unlb}$ [18], as these pseudo-labels are directly linked to the target node importance values in our studied problem. Therefore, let $s_{x'}^+$ denote the constructed pseudo-label for node $x' \in \mathcal{D}'$. The loss term for unlabeled data can be defined as:

$$\mathcal{L}^{unlb} = \frac{1}{|\mathcal{D}'|} \sum_{x' \in \mathcal{D}'} (s_{x'}^+ - \widehat{s}_{x'})^2. \tag{3}$$

## 4 OUR EASING METHODOLOGY

### 4.1 Overview

Despite the intuitiveness of the conventional approach introduced in § 3, one major concern is the supervisory perturbation from unlabeled data, which may introduce the noise that complicates model training by causing deviations in optimization. To address this issue, we propose EASING for effective heterogeneous graph node importance estimation. In this section, we first outline our modifications to the conventional SSL methodology by incorporating the concept of *uncertainty* for unlabeled data measurement in § 4.2. To achieve this, we propose a novel deep neural architecture namely DJE in § 4.3, an encoder-decoder structure designed to estimate both the importance value and uncertainty through *informative distribution modeling*. We then complete our semi-supervised learning paradigm in § 4.4.

### 4.2 Uncertainty-regularized Data Learning

Since leveraging unlabeled data is essential for learning, the supervision signals provided by unlabeled data are usually less influential, compared to those from labeled data. To measure the strength of these supervision signals, we thus consider the concept of "aleatoric uncertainty", which by definition is dependant only on input data [19]. This offers convenience, as we can solely leverage

the input node features to achieve this, without accounting for additional factors. Based on our intuition, the higher uncertainty should contribute less to optimization and vice versa. Inspired by [19], the heteroscedastic loss term for unlabeled data can be formulated:

$$\mathcal{L}^{unlb} = \frac{1}{|\mathcal{D}|} \sum_{x' \in \mathcal{D}'} \frac{(s_{x'}^+ - \widehat{s}_{x'})^2}{2\sigma_{x'}^2} + \frac{\ln \sigma_{x'}^2}{2}, \tag{4}$$

where $\sigma_{x'}$ denotes the uncertainty associated with node $x'$. Eqn. (4) assigns different contributions to data samples with different uncertainty in calculating the loss, i.e., high uncertainty contributes less and vice versa. This differentiates the homoscedastic setting of standard MSE where it essentially assumes equal optimization contribution for all data samples.

As mentioned earlier, since the aleatoric uncertainty only depends on the input data, we can estimate it jointly with the importance value. In this work, we assume that each graph node follows a unique *stochastic distribution*, where the representations of distribution mean and covariance derive the target importance value and uncertainty, respectively. We propose an effective deep neural architecture DJE to perform such joint estimation.

### 4.3 Distribution-based Joint Estimator

Given a heterogeneous graph, we aim to jointly estimate both the node importance values and their associated aleatoric uncertainties via the distribution modeling. Specifically, we use multidimensional *elliptical Gaussian distributions* to represent nodes. An elliptical Gaussian distribution is governed by a mean embedding and a covariance embedding [8], where covariance introduces the associated uncertainty. To achieve this, we propose Distribution-based Joint Estimator (DJE) with an encoder-decoder structure. As shown in Figure 2(A), the encoder captures both the topological and textual features of nodes to represent the informative distributions, while the decoder aggregates the rich encoded information to estimate the importance values and uncertainties.

*4.3.1* **DJE-Encoder.** Given any heterogeneous graph node $x$, $N$ as a hyper-parameter, our DJE-Encoder outputs the mean and covariance representations $S_x, U_x \in \mathbb{R}^{N \times 2d}$ as follows:

$$S_x, U_x = \text{DJE-Encoder}(x). \tag{5}$$

To implement DJE-Encoder, we adopt the Transformer-based structure to process the heterogeneous graph. Specifically, let $H_x^{(0)}$ denote the initial input feature, we start by implementing via a Feed Forward Network (FFN) and Layer Normalization (LN) with residual connection as follows:

$$H_x^{(l+1)} = \text{LN}(\widehat{H}_x^{(l)} + \text{FFN}(\widehat{H}_x^{(l)})), \quad \widehat{H}_x^{(l)} = \text{LN}(H_x^{(l)} + \text{MHA}(H_x^{(l)})). \tag{6}$$

FFN is implemented with two linear layers and ReLU activation. Here MHA denotes the Multi-head attention architecture as follows:

$$\text{MHA}(H_x^{(l)}) = W^{(l)} \Big\|_{h=1}^{H} \text{SHA}^h(H_x^{(l)}). \tag{7}$$

$W^{(l)}$ is the projection matrix and $||$ is the concatenation operation. The $h$-th single-head attention (SHA) is implemented via:

$$\text{SHA}^h(H_x^{(l)}) = \sum_{y \in \mathcal{N}(x)} \alpha_{y \to x}^{(l),h} \cdot W_V^{(l),h} H_x^{(l)}. \tag{8}$$

$\mathcal{N}(x)$ is the neighborhood set of $x$ and $W_V^{(l),h}$ is the $h$-th attentive projection matrix at the $l$-th layer. Then $\alpha_{y \to x}^{(l),h}$ is the $h$-th normalized attention for weighting all connections from node $y$ to $x$:

$$\alpha_{y \to x}^{(l),h} = \sum_{e \in \mathcal{E}(y,x)} \frac{\text{EXP}\left(w_{y \xrightarrow{e} x}^{(l),h}\right)}{\sum_{y' \in \mathcal{N}(x)} \sum_{e' \in \mathcal{E}(y',x)} \text{EXP}\left(w_{y' \xrightarrow{e'} x}^{(l),h}\right)}. \tag{9}$$

$\mathcal{E}(y,x)$ denotes all heterogeneous edges from node $y$ to $x$. $w_{y \xrightarrow{e} x}^{(l),h}$, the unnormalized weight of edge $e$ between nodes $y$ and $x$, is calculated as follows:

$$w_{y \xrightarrow{e} x}^{(l),h} = \frac{W_Q^{(l),h} H_x^{(l)} \cdot \left(W_K^{(l),h} H_y^{(l)}\right)^{\mathsf{T}}}{\sqrt{d}} \cdot W_E^{(l),h} E_{y \xrightarrow{e} x}^{(l)}. \tag{10}$$

$W_Q^{(l),h}$, $W_K^{(l),h}$, and $W_E^{(l),h}$ denote the projection matrix for $h$-th attention at the $l$-th layer. Specifically, $E_{y \xrightarrow{e} x}^{(l)}$ is the trainable embedding for the edge $e$ from node $y$ to $x$.

In this work, we consider both structural and textual features by following [17], and initialize them as $G_x$ and $T_x$ respectively with node2vec [14] and Transformer-XL [10]. Then encoding them with our DJE-Encoder after $L$ iterations, we have the graph node embeddings with concatenation operation, i.e., $H_x = G_x^{(L)} || T_x^{(L)}$. Based on $H_x$, we then derive the distribution mean and covariance representations as follows:

$$S_x = \alpha_s^{\mathsf{T}} \cdot H_x, \quad U_x = \alpha_u^{\mathsf{T}} \cdot H_x, \tag{11}$$

where $\alpha_s$ and $\alpha_u$ are two $N$-dimensional learnable vectors. As both $S_x$ and $U_x$ identify different signals, we then pass them forward for the decoding process in parallel.

*4.3.2* **DJE-Decoder.** Generally, DJE-Decoder estimates the importance value $\widehat{s}_x$ and uncertainty $\widehat{z}_x$ by decoding from the encoded distribution mean and covariance representations as follows:

$$\widehat{s}_x, \widehat{z}_x = \text{DJE-Decoder}(S_x, U_x). \tag{12}$$

Specifically, let $S_x^{(0)}$ and $U_x^{(0)}$ be initialized from Eqn. (11). We iteratively update $S_x^{(l+1)}$ and $U_x^{(l+1)}$ with:

$$S_x^{(l+1)} = \text{LN}(\widehat{S}_x^{(l)} + \text{FFN}_s(\widehat{S}_x^{(l)})), \quad \widehat{S}_x^{(l)} = \text{LN}(S_x^{(l)} + \text{DSA}_s(S_x^{(l)}))$$
$$U_x^{(l+1)} = \text{LN}(\widehat{U}_x^{(l)} + \text{FFN}_u(\widehat{U}_x^{(l)})), \quad \widehat{U}_x^{(l)} = \text{LN}(U_x^{(l)} + \text{DSA}_u(U_x^{(l)})). \tag{13}$$

In Eqn. (13), we implement the adaptive versions of FFN layers for decoding distribution representations as follows:

$$\text{FFN}_s(\widehat{S}_x^{(l)}) = \text{ELU}\left(\text{ELU}(\widehat{S}_x^{(l)} \cdot W_1^s) W_2^s\right),$$
$$\text{FFN}_u(\widehat{S}_x^{(l)}) = \text{ELU}\left(\text{ELU}(\widehat{S}_x^{(l)} \cdot W_1^u) W_2^u\right), \tag{14}$$

where we use $\text{ELU}(\cdot)$, the exponential linear unit activation, particularly for numerical stability in decoding [7]. $W_1^s$, $W_2^s$ are two transformation matrices. In Eqn. (13), DSA denotes our *distribution-based self-attention*, which is defined as:

$$\text{DSA}_s(S_x^{(l)}) = \text{SOFTMAX}\left(\frac{Q_x^{s(l)} K_x^{s(l)\mathsf{T}}}{\sqrt{d}}\right) \cdot V_x^{s(l)},$$
$$\text{DSA}_u(U_x^{(l)}) = \text{SOFTMAX}\left(\frac{Q_x^{u(l)} K_x^{u(l)\mathsf{T}}}{\sqrt{d}}\right) \cdot V_x^{u(l)}, \tag{15}$$

where we apply the transformations with $W_Q^{(l)}, W_K^{(l)}, W_V^{(l)} \in \mathbb{R}^{2d \times 2d}$ to equip with extra non-linearity from $\text{ELU}(\cdot)$ activation:

$$Q_x^{s(l)} = \text{ELU}(S_x^{(l)} W_Q^{(l)}), \quad Q_x^{u(l)} = \text{ELU}(U_x^{(l)} W_Q^{(l)}),$$
$$K_x^{s(l)} = \text{ELU}(S_x^{(l)} W_K^{(l)}), \quad K_x^{u(l)} = \text{ELU}(U_x^{(l)} W_K^{(l)}), \tag{16}$$
$$V_x^{s(l)} = \text{ELU}(S_x^{(l)} W_V^{(l)}), \quad V_x^{u(l)} = \text{ELU}(U_x^{(l)} W_V^{(l)}).$$

After $L$ layers of iteration, we output the following estimation by conducting the Frobenius inner product with two learnable matrices $W_s, W_z \in \mathbb{R}^{N \times 2d}$:

$$\widehat{s}_x = \langle W_s, S_x^{(L)} \rangle_F, \quad \widehat{z}_x = \langle W_z, U_x^{(L)} \rangle_F. \tag{17}$$

$\widehat{s}_x$ and $\widehat{z}_x$ denote the estimated importance values and uncertainty of the input node $x$. In this work, we also introduce an auxiliary scaling trick in implementation to adjust the importance value ranges. Due to page limits, we report the details in Appendix A. Utilizing these estimates, we proceed to finalize our semi-supervised learning approach as described below.

## 4.4 Semi-supervised Heteroscedastic Regression

*4.4.1* **Pseudo-label Generation.** For unlabeled nodes $\mathcal{D}' = \{x'\}$, ground-truth importance is unknown, which makes the prediction challenging. To handle this, We aim to obtain the pseudo-labels for node $x'$ in $\mathcal{D}'$, i.e., the importance value $s_{x'}^+$ and uncertainty $z_{x'}^+$. In this work, we adopt *ensembling techniques with variational inference* directly from DJE to reduce the predictor error. Concretely, our DJE is implemented with Monte Carlo dropout in its architecture, which is a common approach used in Bayesian deep learning for variational inference [11]. We create two independent DJE structures, denoted by DJE$_1$ and DJE$_2$, with the variational estimation produced as follows:

$$\widehat{s}_{x',1}, \widehat{z}_{x',1} = \text{DJE}_1(x'), \quad \widehat{s}_{x',2}, \widehat{z}_{x',2} = \text{DJE}_2(x'). \tag{18}$$

Then based on $T$ times of predictions for ensembling, where $\widehat{s}_{x',i}^{<t>}$ denotes the $t$-th individual prediction from Eqn. (18), the pseudo-labels thus can be constructed as follows:

$$s_{x'}^+ = \frac{1}{2T} \sum_{t=1}^{T} \sum_{i=1}^{2} \widehat{s}_{x',i}^{<t>}, \quad z_{x'}^+ = \frac{1}{2T} \sum_{t=1}^{T} \sum_{i=1}^{2} \widehat{z}_{x',i}^{<t>}. \quad (19)$$

As shown in Theorem 1, our ensembled pseudo-labels will always have smaller expected discrepancy than each individual, i.e., $\widehat{s}_{x',i}^{<t>}$, compared to the "hidden ground-truth" label $s_{x'}$ as the target.

THEOREM 1 (PSEUDO-LABEL QUALITY). *For $x' \in \mathcal{D}'$ and $i \in \{1,2\}$, let $s_{x'}$ be the "hidden ground-truth" label of $x'$ and we have:*

$$E[(s_{x'} - \widehat{s}_{x',i})^2] \geq E[(s_{x'} - s_{x'}^+)^2]. \quad (20)$$

PROOFS OF THEOREM 1. We apply the bias-covariance decomposition as follows:

$$E[(s_{x'} - \widehat{s}_{x',i})^2] = \underbrace{(s_{x'} - E[\widehat{s}_{x',i}])^2}_{\text{Bias of: } \widehat{s}_{x',i}} + \underbrace{E[(\widehat{s}_{x',i} - E[\widehat{s}_{x',i}])^2]}_{\text{covariance of: } \widehat{s}_{x',i}},$$

$$E[(s_{x'} - s_{x'}^+)^2] = \underbrace{(s_{x'} - E[s_{x'}^+])^2}_{\text{Bias of: } s_{x'}^+} + \underbrace{E[(s_{x'}^+ - E[s_{x'}^+])^2]}_{\text{covariance of: } s_{x'}^+}. \quad (21)$$

Since $s_{x'}^+ = \frac{1}{2T} \sum_{t=1}^{T} \sum_{i=1}^{2} \widehat{s}_{x',i}^{<t>}$, for the bias term $(s_{x'} - E[s_{x'}^+])^2$:

$$(s_{x'} - E[s_{x'}^+])^2 = (s_{x'} - E[\frac{1}{2T} \sum_{t=1}^{T} \sum_{i=1}^{2} \widehat{s}_{x',i}^{<t>}])^2$$

$$= (s_{x'} - \frac{1}{2T} \sum_{t=1}^{T} \sum_{i=1}^{2} E[\widehat{s}_{x',i}^{<t>}])^2 \quad (22)$$

$$= \underbrace{(s_{x'} - E[\widehat{s}_{x',i}])^2}_{\text{Bias of: } \widehat{s}_{x',i}}.$$

For the covariance term $s_{x'}^+$, similarly we have:

$$E[(s_{x'}^+ - E[s_{x'}^+])^2] = Var(s_{x'}^+) = Var(\frac{1}{2T} \sum_{t=1}^{T} \sum_{i=1}^{2} \widehat{s}_{x',i}^{<t>})$$

$$= \frac{1}{4T^2} \sum_{t=1}^{T} \sum_{i=1}^{2} Var(\widehat{s}_{x',i}^{<t>})$$

$$= \frac{1}{4T} \sum_{i=1}^{2} \underbrace{\sum_{t=1}^{T} Var(\widehat{s}_{x',i})}_{\text{covariance of: } \widehat{s}_{x',i}} \quad (23)$$

$$\leq E[(\widehat{s}_{x',i} - E[\widehat{s}_{x',i}])^2],$$

which completes the proof. $\square$

### 4.4.2 *Semi-supervised Heteroscedastic Learning Objective.*
For unlabeled nodes $\mathcal{D}' = \{(x')\}$, we rewrite their regression loss based on the generated pseudo labels as follows:

$$\mathcal{L}_{reg}^{unlb} = \frac{1}{|\mathcal{D}'|} \sum_{x' \in \mathcal{D}'} \sum_{i=1}^{2} \left( \frac{(s_{x'}^+ - \widehat{s}_{x',i})^2}{2 \operatorname{EXP}(\widehat{z}_{x',i})} + \frac{\widehat{z}_{x',i}}{2} \right). \quad (24)$$

In Eqn. (24), we use $\operatorname{EXP}(\widehat{z}_{x',i})$, i.e., meaning that $\widehat{z}_{x',i} = \ln \sigma^2$, mainly for numerical stabilization. In addition, to further reduce

**Table 1: Dataset statistics.**

|  | FB15K | TMDB5K | IMDB |
|---|---|---|---|
| # Edge | 592,213 | 761,648 | 1,123,808 |
| # Node | 14,951 | 114,805 | 150,000 |
| # Edge type | 1,345 | 34 | 30 |
| # Training labeled nodes | 1,407 | 479 | 4,841 |
| # Training label ratio | 9.41% | 0.42% | 3.23% |

the potential perturbation caused by the inaccurate estimation, we regularize our model to stabilize uncertainty estimation:

$$\mathcal{L}_{stab}^{unlb} = \frac{1}{|\mathcal{D}'|} \sum_{x' \in \mathcal{D}'} \sum_{i=1}^{2} \left( z_{x'}^+ - \widehat{z}_{x',i} \right)^2. \quad (25)$$

Therefore, the objective for unlabeled data is formally defined:

$$\mathcal{L}^{unlb} = \mathcal{L}_{reg}^{unlb} + \mathcal{L}_{stab}^{unlb}. \quad (26)$$

For labeled data $\mathcal{D} = \{(x, s_x)\}$, please notice that, we also upgrade the original homoscedastic formulation in Eqn. (2) by the uncertainty-regularized heteroscedastic setting as follows:

$$\mathcal{L}_{reg}^{lb} = \frac{1}{|\mathcal{D}|} \sum_{x \in \mathcal{D}} \frac{(s_x - \widehat{s}_x)^2}{2 \operatorname{EXP}(\widehat{z}_x)} + \frac{\widehat{z}_x}{2}. \quad (27)$$

This occurs because it provides a weak regularization effect on our uncertainty estimation, as optimizing the model on labeled data tends to reduce uncertainty more effectively compared to unlabeled data. This is validated in § 5.3, thereby demonstrating better performance than the homoscedastic setting. Additionally, we also minimize the uncertainty estimation disagreement:

$$\mathcal{L}_{stab}^{lb} = \frac{1}{|\mathcal{D}|} \sum_{x \in \mathcal{D}} (\widehat{z}_{x,1} - \widehat{z}_{x,2})^2. \quad (28)$$

Finally, with the illustration shown in Figure 2(B), we complete our semi-supervised heteroscedastic objective function as follows:

$$\mathcal{L} = \mathcal{L}_{reg}^{lb} + \mathcal{L}_{stab}^{lb} + \lambda(\mathcal{L}_{reg}^{unlb} + \mathcal{L}_{reg}^{unlb}). \quad (29)$$

To summarize, we optimize our model for labeled and unlabeled data with node distribution modeling and uncertainty regularization. For unlabeled data, we leverage variational model ensembling to explicitly construct high-quality pseudo-labels, which provide effective supervisory signals in model optimization and eventually produce accurate estimation of node importance values.

## 5 EXPERIMENTS
We evaluate our EASING with the aim of answering the following research questions:

- **RQ1**: How does EASING perform on the real-world data, compared to state-of-the-art methods, on the tasks of (1) *node importance value estimation* and (2) *node importance ranking*?
- **RQ2**: How does our uncertainty regularization benefit EASING?
- **RQ3**: How does each proposed module component of EASING contribute to the model performance?
- **RQ4**: How can we further evaluate EASING, from other aspects of scalability, model compatibility, and hyper-parameter sensitivity?

**Table 2: (1) Overall performance on node importance value estimation task; (2) notation * denotes the case that the performance improvement of EASING is significant with $p$-value less than 0.05; (3) we use underline to denote the best-performing baseline models and use bold to denote the case where our model achieves better performance.**

| Dataset | FB15K | | | TMDB5K | | | IMDB | | |
|---|---|---|---|---|---|---|---|---|---|
| Method | MAE | RMSE | NRMSE | MAE | RMSE | NRMSE | MAE | RMSE | NRMSE |
| PR | 10.0034 ± 0.026 | 10.1116 ± 0.025 | 0.9910 ± 0.118 | 2.5287 ± 0.036 | 2.7787 ± 0.036 | 0.4743 ± 0.051 | 6.0401 ± 0.036 | 6.6465 ± 0.038 | 0.5332 ± 0.008 |
| PPR | 10.0034 ± 0.026 | 10.1116 ± 0.025 | 0.9905 ± 0.118 | 2.5286 ± 0.036 | 2.7786 ± 0.036 | 0.4743 ± 0.051 | 6.0401 ± 0.036 | 6.6465 ± 0.038 | 0.5331 ± 0.008 |
| LR | 1.5037 ± 0.040 | 2.1070 ± 0.083 | 0.2069 ± 0.029 | 1.4138 ± 0.085 | 2.0377 ± 0.097 | 0.3476 ± 0.038 | 2.0473 ± 0.022 | 2.6033 ± 0.037 | 0.2088 ± 0.002 |
| RF | 0.9691 ± 0.013 | 1.2440 ± 0.020 | 0.1217 ± 0.014 | 0.6482 ± 0.019 | 0.8148 ± 0.031 | 0.1386 ± 0.011 | 1.8415 ± 0.015 | 2.2232 ± 0.019 | 0.1783 ± 0.003 |
| GENI | 1.1406 ± 0.134 | 1.5079 ± 0.128 | 0.1483 ± 0.026 | 0.7551 ± 0.180 | 0.9298 ± 0.198 | 0.1569 ± 0.027 | 1.3283 ± 0.045 | 1.6729 ± 0.056 | 0.1342 ± 0.005 |
| MULTI | 0.8743 ± 0.072 | 1.2394 ± 0.093 | 0.1249 ± 0.088 | 0.7083 ± 0.102 | 0.8693 ± 0.093 | 0.1445 ± 0.078 | 1.1952 ± 0.048 | 1.5932 ± 0.034 | 0.1233 ± 0.049 |
| RGTN | 0.7981 ± 0.034 | 1.0586 ± 0.049 | 0.1035 ± 0.012 | 0.6459 ± 0.011 | 0.8069 ± 0.016 | 0.1376 ± 0.014 | 1.1912 ± 0.037 | 1.5330 ± 0.044 | 0.1230 ± 0.004 |
| HIVEN | 0.8418 ± 0.084 | 1.1937 ± 0.049 | 0.1134 ± 0.077 | 0.6645 ± 0.093 | 0.8180 ± 0.058 | 0.1377 ± 0.055 | 1.1831 ± 0.037 | 1.5644 ± 0.062 | 0.1240 ± 0.090 |
| SKES | 0.7745 ± 0.029 | 1.0248 ± 0.045 | 0.1043 ± 0.022 | 0.6488 ± 0.020 | 0.8021 ± 0.034 | 0.1403 ± 0.038 | 1.1834 ± 0.042 | 1.5229 ± 0.044 | 0.1225 ± 0.023 |
| UBDL | 1.3752 ± 0.235 | 1.6821 ± 0.240 | 0.1667 ± 0.040 | 1.0115 ± 0.103 | 1.2748 ± 0.150 | 0.2179 ± 0.036 | 2.3351 ± 0.080 | 2.7979 ± 0.046 | 0.2245 ± 0.006 |
| SSDPKL | 0.8959 ± 0.007 | 1.1554 ± 0.008 | 0.1072 ± 0.001 | 0.7580 ± 0.016 | 0.9396 ± 0.018 | 0.1505 ± 0.003 | 1.7325 ± 0.010 | 2.1464 ± 0.005 | 0.1704 ± 0.000 |
| UCVME | 0.9197 ± 0.007 | 1.1890 ± 0.010 | 0.1164 ± 0.014 | 0.7267 ± 0.027 | 0.9092 ± 0.019 | 0.1550 ± 0.015 | 1.9563 ± 0.017 | 2.3589 ± 0.019 | 0.1892 ± 0.002 |
| **EASING** | **0.7315** ± 0.017 | **0.9846** ± 0.023 | **0.0964** ± 0.012 | **0.6226** ± 0.012 | **0.7760** ± 0.012 | **0.1324** ± 0.014 | **1.1402** ± 0.008 | **1.4494** ± 0.005 | **0.1163** ± 0.002 |
| Gain | +5.55%* | +3.92%* | +6.86%* | +3.61%* | +3.25%* | +3.78%* | +3.63%* | +4.83%* | +5.06%* |

## 5.1 Setups

*5.1.1 **Datasets**.* We include three widely evaluated real-world heterogeneous graphs, namely FB15K, TMDB5K, and IMDB. For a fair comparison, we follow the datasets processed in [17]. Dataset statistics are reported in Table 1 with descriptions as follows. Please notice that, in this work, the training data of all these datasets share the same size with validation and testing data.

- **FB15K** is processed as a subset from the FreeBase [1]. It contains rich heterogeneous knowledge information including both relational and textual information [3]. Specifically, the textual information is derived from the wikidata description. The labeled node importance is based on the pageview number in the last 30 days of the corresponding Wikipedia page.
- **TMDB5K** is originated from the movie database TMDB[1] and built on public data[2]. It includes heterogeneous nodes such as actors, casts, crews, and companies. The semantic information is based on movie overviews and the node importance is labeled based on the official movie popularity score.
- **IMDB** is processed from the IMDB database[3], which contains the heterogeneous graph nodes including movies, genres, casts, crews, publication companies, and countries. The textual information of this dataset is from the IMDB movie summaries and personal biographies. The labeled node importance is derived from the IMDB movie vote number.

*5.1.2 **Competing Methods**.* We include three categories of existing methods for comparison: (1) traditional network analytic methods, i.e., PageRank (PR) [24] and personalized PageRank (PPR) [15]; (2) supervised machine learning methods, i.e., linear regression (LR) and random forest (RF), and supervised GNN-based models, i.e., GENI [25], Multiimport (MULTI) [26], RGTN [17], HIVEN [16], and SKES [5]. (4) semi-supervised machine-learning-based regression methods, i.e., UBDL [19], SSDPKL [20], and UCVME [9].

- **PR** [24] is a classic algorithm based on the random walk for node importance estimation and **PPR** [15] is a variant of PageRank biased by relevant importance values.
- **LR** is one of the classic machine learning methods using a least squares algorithm to minimize prediction errors of regression.
- **RF** is an ensemble learning method for regression using bagging techniques.
- **GENI** [25] is a GNN model that heterogeneously aggregates neighbor importance values considering edge types, and adjusts estimation by node centrality.
- **MULTI** [26] is an improved version of GENI generating unified importance values for each node from heterogeneous inputs.
- **RGTN** [17] is a representative GNN-based model that jointly considers both topological and textual information for node importance estimation.
- **HIVEN** [16] is one of the latest heterogeneous GNN models capturing both local and global information to improve node importance estimation accuracy.
- **SKES** [5] is the latest GNN model that leverages transformer and optimal transport theory for node important estimation.
- **UBDL** [19] is a representative Bayesian deep learning framework to quantify uncertainty for regression tasks.
- **SSDPKL** [20] is a classic probabilistic neural network that learns deep kernels for semi-supervised setting.
- **UCVME** [9] is the latest semi-supervised learning model with uncertainty-consistent awareness for deep regression.

*5.1.3 **Evaluation Metrics**.* For comprehensive evaluation, we follow previous work [5, 17] to introduce two evaluation tasks: (1) *node importance value estimation task* and (2) *node importance ranking task*. For the node importance value estimation task, three metrics are applied for performance evaluation, including mean absolute error (MAE), root mean square error (RMSE), and normalized root mean square error (NRMSE). The lower the value of these metrics, the better the model performance. On the importance ranking task, we use Spearman correlation coefficient (SPEARMAN), Precision, and normalized discounted cumulative gain (NDCG), where a higher value indicates better performance.

---

[1]https://www.themoviedb.org/
[2]https://www.kaggle.com/tmdb/tmdb-movie-metadata
[3]https://www.imdb.com/interfaces/

**Table 3: Uncertainty regularization for labeled data.**

| | FB15K | | TMDB5K | | IMDB | |
|---|---|---|---|---|---|---|
| | NRMSE | SPEARMAN | NRMSE | SPEARMAN | NRMSE | SPEARMAN |
| Homoscedastic $\mathcal{L}^{lb}$ | 0.0965 | 0.7497 | 0.1498 | 0.7380 | 0.1220 | 0.7841 |
| | ↓0.10% | ↓0.29% | ↓13.11% | ↓2.22% | ↓4.94% | ↓1.60% |
| **EASING** | **0.0964** | **0.7519** | **0.1324** | **0.7547** | **0.1163** | **0.7968** |

*5.1.4* ***Experimental Settings***. In line with prior work [16, 25], we perform five-time evaluation and report the average performance. EASING is implemented with Python 3.8 and PyTorch 1.8.1 on a Linux machine with 4 Nvidia RTX 3090 GPUs and 28 14-core 45GB Intel(R) Xeon(R) Platinum 8362 CPUs @ 2.80GHz. For all competing models, we either directly follow their official parameter settings, or apply grid search if the settings are not available. We set the learning rate as $5 \times 10^{-3}$. We train the model via Adam optimizer and report hyper-parameter settings in Appendix B.2.

## 5.2 Overall Performance (RQ1)

*5.2.1* ***Task of Importance Value Estimation***. We first evaluate our EASING model on the task of importance value estimation. As shown in Table 2, we have the following fourfold observations. (1) Compared to the traditional graph analytical methods, i.e., PR and PPR, learning-based approaches generally consistently present better performance over them. This shows the efficacy of utilizing supervision signals, i.e., from either supervised or semi-supervised learning paradigms, in improving the estimation accuracy. (2) Compared to the general machine learning methods, i.e., LR and RF, deep-learning-based methods that are specifically designed for node importance estimation further show the general effectiveness, which is reasonable as they are good at aggregating graph information for regression. (3) However, the three semi-supervised methods, i.e., UBDL, SSDPKL, and UCVME, underperform the supervised methods on this task, as they are more general approaches but not particularly designed for graph-based regression. This highlights the importance of developing a semi-supervised methodology, especially for graph-based importance estimation problems. (4) Our EASING model showcases the consistent performance superiority over existing works, with performance gain from 3.25% to 6.86%, demonstrating its effectiveness of leveraging heterogeneous graph information and uncertainty measurement in semi-supervised learning paradigm for node importance estimation tasks. In addition, we conduct the Wilcoxon signed-rank tests to show that all the performance improvements are statistically significant with at least a 95% confidence level.

*5.2.2* ***Task of Importance Ranking***. As for this task, it is inherently correlated to the task of node importance value estimation. Due to page limits, we report the complete results and analyses in Appendix B.1. Generally, among all these methods, EASING continues to demonstrate superior performance over other competing methods via achieving 0.22% to 5.94% performance improvement.

## 5.3 Study of Uncertainty Regularization (RQ2)

To evaluate the effectiveness of our design in considering uncertainty, we introduce the following three experiments.

(1) We firstly disable the uncertainty-regularized learning for label data, i.e., Eqn.'s (27-28), by replacing it with the homoscedastic one, i.e., Eqn. (2). The results in Table 3 empirically demonstrate

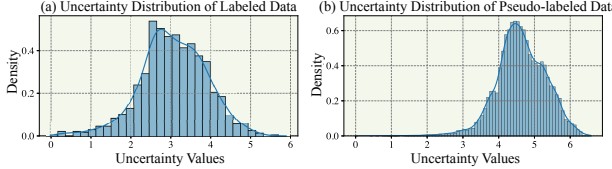

**Figure 3: Uncertainties of labeled and pseudo-labeled data.**

**Table 4: Uncertainty estimation comparison on FB15K.**

| | RMSE | NRMSE | Precision | NDCG | Inference Time |
|---|---|---|---|---|---|
| EASING$_{BNN}$ | 1.0078 | 0.0989 | 0.4440 | 0.9430 | 0.4118 (s) |
| **EASING** | 0.9846 | 0.0964 | 0.4780 | 0.9506 | 0.4793 (s) |

**Table 5: Ablation study of EASING on FB15K dataset.**

| | MAE | RMSE | NRMSE | SPEARMAN | Precision | NDCG |
|---|---|---|---|---|---|---|
| w/o DJE-Encoder | 1.5601 | 1.8760 | 0.1928 | 0.5148 | 0.3740 | 0.9131 |
| | ↓≥100% | ↓90.53% | ↓≥100% | ↓31.53% | ↓21.76% | ↓3.94% |
| w/o DJE-Decoder | 0.7558 | 1.0044 | 0.0984 | 0.7340 | 0.4460 | 0.9429 |
| | ↓3.32% | ↓2.01% | ↓2.07% | ↓2.38% | ↓6.69% | ↓0.81% |
| w/o SSIL | 0.7606 | 1.0268 | 0.1005 | 0.7274 | 0.4840 | 0.9459 |
| | ↓3.98% | ↓4.29% | ↓4.23% | ↓3.26% | ↑1.26% | ↓0.49% |
| w/o AST | 0.7522 | 1.010 | 0.0987 | 0.7318 | 0.4660 | 0.9429 |
| | ↓2.83% | ↓2.58% | ↓2.39% | ↓2.67% | ↓2.51% | ↓0.81% |
| **EASING** | **0.7315** | **0.9846** | **0.0964** | **0.7519** | **0.4780** | **0.9506** |

that incorporating uncertainty regularization into labeled data learning, as additional supervision signals, is beneficial.

(2) We then visualize the estimated uncertainty values of both labeled and pseudo-labeled data in Figure 3. We observe that, for labeled data, our model can generally have a lower uncertainty measurement with the major values in [2.5, 4]; on the contrary, for pseudo-labeled data, the majority of uncertainty values are in around [4, 5.5]. Such a difference makes sense in practice, as we expect the model to have more confidence in the labeled data, and present flexibility to pseudo-labeled data.

(3) We further compare DJE with another common structure, i.e., Bayesian neural networks (BNNs), for uncertainty estimation. Specifically, we replace our DJE-Decoder by implementing the BNN with a two-layer MLP and random dropout. We report the results in Table 4 including four accuracy metrics and average time cost per epoch for inferring the uncertainty. With similar inference time costs, our original design achieves better performance than using simple BNNs. This shows that the deeper structure in our DJE-Decoder is more effective in measuring the uncertainty and optimizing our model accordingly.

## 5.4 Ablation Study (RQ3)

To investigate the effectiveness of each proposed module in EASING, we construct four variants by disabled their functionality for ablation study. We report the results of experimenting on FB15K dataset in Table 5 and provide the analyses accordingly.

(1) w/o DJE-Encoder: We first disable our DJE-Encoder implementation and directly replace mapping the initial features for distribution modeling. The variant of w/o DJE-Encoder presents a huge performance decay, e.g., with over 90.53% and 3.94% in importance estimation and ranking tasks, respectively, compared to our complete version. This implies the criticalness of such

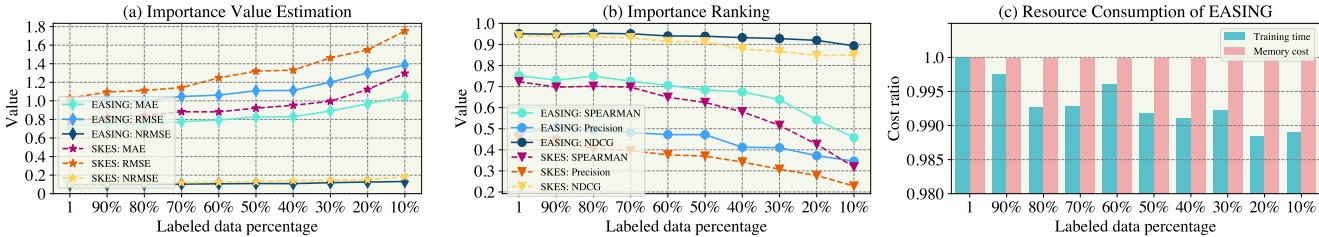

**Figure 4: Model performance with different labeled data percentage.**

**Table 6: Performance of EASING-enhanced models on FB15K.**

| Method | RMSE | NRMSE | Precision | NDCG |
|---|---|---|---|---|
| RGTN | 1.0586 | 0.1035 | 0.4480 | 0.9396 |
| EASING→ RGTN | 1.0227 (↑3.51%) | 0.1001 (↑3.40%) | 0.4560 (↑1.75%) | 0.9432 (↑0.38%) |
| SKES | 1.0248 | 0.1043 | 0.4512 | 0.9411 |
| EASING→ SKES | 1.0103 (↑1.41%) | 0.1020 (↑2.21%) | 0.4627 (↑2.55%) | 0.9486 (↑0.80%) |

module design in encoding the distribution representations from the graph topological and textual features.

(2) `w/o DJE-Decoder`: We further replace our `DJE-Decoder` module with a single linear transformation for outputting the importance values and uncertainties. Our experimental results generally demonstrate the effectiveness of `DJE-Decoder` module in uncertainty measurement.

(3) `w/o SSIL`: To evaluate our semi-supervised importance learning objectives, we construct the variant namely w/o SSIL by canceling the pseudo-label usage and unlabeled data learning. As shown in Table 5, the performance gap between w/o SSIL and EASING basically proves that our learning of unlabeled data is essential to improve the prediction accuracy, particularly in the semi-supervised learning settings.

(4) `w/o AST`: Lastly, we study our auxiliary scaling trick by simply removing it in the variant w/o AST. On one hand, the experimental results showcase its usefulness in further enhancing the model prediction capability. On the other hand, since it requires additional processing, we therefore take it as an alternative with the consideration of the efficiency-effectiveness tradeoff.

## 5.5 Empirical Analyses of EASING (RQ4)

*5.5.1 **Scalability with Varying Labeled Data Ratio**.* (1) To study our model's semi-supervised learning capability, we first introduce the latest supervised model SKES [5], i.e., only trained by the labeled data; and then we thoroughly evaluate the performance of EASING and SKES, by subsequently varying the labeled data ratio from 100%, i.e., the complete label data for training, to 10%, i.e., the random sample of the labeled data. We present the metrics of estimation accuracy for two models in Figure 4 (a) and (b). With our proposed learning design for unlabeled data, i.e., pseudo-label generation and utilization, EASING presents a less decayed performance trend, compared to the supervised counterpart SKES. This demonstrates the effectiveness of our semi-supervised learning paradigm in leveraging unlabeled data for enhancing model robustness. (2) Furthermore, we report EASING's training time cost/per epoch and GPU memory usage in the cases of *with or without unlabeled data usage*. Notice that, due to the different scales of time and

memory cost, we set the initial state as 1 and compute their value gaps accordingly for ease of illustration in Figure 4(c). We observe that, with label data being reduced, EASING presents gradually decreasing training time as both labeled and pseudo-labeled data are scaled down. In addition, the GPU memory usage is generally stable, indicating that the space cost of EASING is not directly determined by the data size. Overall, these observations may discern the concerns of heavy computation for unlabeled data.

*5.5.2 **Compatibility to Other Models**.* We apply our proposed semi-supervised importance learning framework, to other existing methods, e.g., RGTN and SKES. Concretely, we keep their embedding learning parts unchanged, and increment the pseudo-label generation as well as the uncertainty-aware regression into their proposed methods. From the results in Table 6, both two models achieve a competitive performance improvement, verifying that our proposed semi-supervised learning framework provides good compatibility to boost existing models.

*5.5.3 **Hyper-parameter Sensitivity**.* Finally, we investigate how model performance varies under different hyper-parameter settings. The detailed analyses can be found in Appendix B.3.

## 6 CONCLUSIONS AND FUTURE WORK

In this work, we propose the framework EASING to study the problem of heterogeneous graph node importance estimation in the semi-supervised learning setting. EASING considers measuring uncertainty alongside node importance estimation for regularization of different data in model optimization. EASING introduces `DJE`, an encoder-decoder architecture to encode rich heterogeneous graph information for node distribution modeling, followed by the decoding of distribution information for joint estimation of node importance values and uncertainty. `DJE` facilitates the construction of high-quality pseudo labels for initially unlabeled data, thereby enriching the dataset for model training. The empirical analyses demonstrate the effectiveness of our proposed methods on three extensively studied real-world datasets.

For future work, we plan to explore (1) extending our method to temporal graphs, with a focus on capturing the *evolving nature of importance values* across different timestamps, and (2) adapting our approach to other domains, such as medical discovery, where estimating the importance of key factors in medical knowledge graphs could provide valuable insights.

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

# A    SUPPLEMENTARIES OF AUXILIARY SCALING TRICK

As pointed out by [5, 16], the graph node importance values may have strong correlations with the topological centralities, e.g., Eigenvector centrality [2]. Inspired by this, we introduce the following scaling trick to further enhance the estimation performance. Concretely, we consider two types of auxiliary information, i.e., *graph structure centrality* and *accumulative textual relative entropy*.

- **Graph structure centrality.** Given any node $x$, we compute its logarithm degree centrality as follows:

$$\text{CEN}(x) = \log\left(|\mathcal{N}(x)| + \delta\right), \tag{30}$$

where $\delta$ is a small term, e.g., $\delta = 1e^{-4}$.

- **Accumulative texutal relative entropy.** The $d$-dimensional textual features $T_x$ encoded by Transformer-XL are firstly normalized with $Q_x = \text{SOFTMAX}(T_x)$. For normalized textual feature $Q_x$, we use $Q_x(j)$ to denote its $j$-th element. Then $x$'s textual relative entropy is calculated as:

$$\text{RENT}(x) = \log\left(\sum_{y \in \mathcal{N}(x)} \sum_{j=1}^{d} Q_x(j) \log \frac{Q_x(j)}{Q_y(j)} + \delta\right). \tag{31}$$

Then we directly include these two scalers to update the estimated importance values, i.e., computed in Eqn. (17), as follows and leave the uncertainty unchanged.

$$\widehat{s}_x = \left(\mu_1 \text{CEN}(x) + \mu_2 \text{RENT}(x)\right) \cdot \widehat{s}_x, \text{ where } \mu_1, \mu_2 \in (0,1). \tag{32}$$

# B    SUPPLEMENTARIES OF EXPERIMENTS

## B.1    Experimental Results for Task of Importance Ranking

We report the evaluation results of the importance ranking task in Table 7. (1) We observe that, the existing methods present similar performance trends with those in the value estimation tasks. This is reasonable as the capability of quantifying the node importance values is inherently correlated to the importance-based ranking. Among all these methods, EASING continues to demonstrate superior performance over other competing methods via achieving 0.22% to 5.94% performance improvement. (2) Furthermore, as mentioned by [5, 17], one possible solution to improve the model ranking capability is to incorporate the ranking objective. To validate this,

**Table 7: (1) Overall performance on node importance ranking task; (2) notation * denotes the case that the performance improvement of EASING is significant with $p$-value less than 0.05; (3) we use underline to denote the best-performing baseline models and use bold to denote the case where our model achieves better performance.**

| Dataset | FB15K | | | TMDB5K | | | IMDB | | |
|---|---|---|---|---|---|---|---|---|---|
| Method | SPEARMAN | Precision@100 | NDCG@100 | SPEARMAN | Precision@100 | NDCG@100 | SPEARMAN | Precision@100 | NDCG@100 |
| PR | 0.3629 ± 0.014 | 0.2140 ± 0.023 | 0.8408 ± 0.013 | 0.6210 ± 0.016 | 0.5300 ± 0.023 | 0.8556 ± 0.017 | 0.5609 ± 0.006 | 0.3820 ± 0.052 | 0.8764 ± 0.020 |
| PPR | 0.3602 ± 0.016 | 0.2140 ± 0.021 | 0.8419 ± 0.014 | 0.7190 ± 0.020 | 0.5640 ± 0.021 | 0.8767 ± 0.013 | 0.6482 ± 0.006 | 0.4440 ± 0.055 | 0.9097 ± 0.014 |
| LR | 0.4258 ± 0.012 | 0.1780 ± 0.046 | 0.8467 ± 0.014 | 0.3859 ± 0.013 | 0.3940 ± 0.034 | 0.7337 ± 0.018 | 0.4971 ± 0.010 | 0.1960 ± 0.037 | 0.6338 ± 0.032 |
| RF | 0.5623 ± 0.035 | 0.3240 ± 0.024 | 0.9088 ± 0.010 | 0.6890 ± 0.013 | 0.5440 ± 0.029 | 0.8604 ± 0.016 | 0.5177 ± 0.006 | 0.3340 ± 0.036 | 0.8798 ± 0.009 |
| GENI | 0.6344 ± 0.029 | 0.3640 ± 0.041 | 0.9028 ± 0.004 | 0.7090 ± 0.027 | 0.5960 ± 0.054 | 0.8876 ± 0.037 | 0.7376 ± 0.013 | 0.5840 ± 0.027 | 0.9572 ± 0.008 |
| MULTI | 0.6838 ± 0.034 | 0.3939 ± 0.044 | 0.9133 ± 0.012 | 0.7322 ± 0.039 | 0.5973 ± 0.073 | 0.8949 ± 0.042 | 0.7589 ± 0.028 | 0.6044 ± 0.033 | 0.9533 ± 0.019 |
| RGTN | 0.7171 ± 0.025 | 0.4480 ± 0.026 | 0.9396 ± 0.006 | 0.7560 ± 0.019 | 0.6160 ± 0.034 | 0.9021 ± 0.018 | 0.7864 ± 0.006 | 0.6000 ± 0.034 | 0.9629 ± 0.004 |
| HIVEN | 0.7145 ± 0.033 | 0.4497 ± 0.083 | 0.9338 ± 0.027 | 0.7419 ± 0.043 | 0.5987 ± 0.079 | 0.8918 ± 0.024 | 0.7723 ± 0.033 | 0.5934 ± 0.045 | 0.9590 ± 0.022 |
| SKES | 0.7234 ± 0.030 | 0.4512 ± 0.034 | 0.9411 ± 0.013 | 0.7544 ± 0.021 | 0.6144 ± 0.054 | 0.9008 ± 0.029 | 0.7873 ± 0.027 | 0.5943 ± 0.045 | 0.9618 ± 0.039 |
| UBDL | 0.1650 ± 0.035 | 0.1040 ± 0.023 | 0.8131 ± 0.019 | 0.2580 ± 0.147 | 0.3520 ± 0.096 | 0.7086 ± 0.086 | 0.1566 ± 0.044 | 0.0200 ± 0.013 | 0.4553 ± 0.055 |
| SSDPKL | 0.6345 ± 0.008 | 0.3000 ± 0.017 | 0.9204 ± 0.004 | 0.6331 ± 0.016 | 0.5880 ± 0.004 | 0.8947 ± 0.006 | 0.5683 ± 0.002 | 0.2860 ± 0.033 | 0.8808 ± 0.015 |
| UCVME | 0.6131 ± 0.016 | 0.3020 ± 0.050 | 0.8992 ± 0.006 | 0.6323 ± 0.025 | 0.5720 ± 0.021 | 0.8781 ± 0.007 | 0.5126 ± 0.015 | 0.2700 ± 0.059 | 0.7997 ± 0.017 |
| **EASING** | **0.7519** ± 0.016 | **0.4780** ± 0.049 | **0.9506** ± 0.008 | 0.7547 ± 0.024 | 0.6040 ± 0.030 | **0.9058** ± 0.002 | **0.7968** ± 0.015 | **0.6100** ± 0.026 | **0.9650** ± 0.003 |
| Gain | +3.94%* | +5.94%* | +1.01%* | - | - | +0.41%* | +1.21%* | +0.93%* | +0.22% |
| **EASING$_{rank}$** | **0.7489** ± 0.010 | **0.5020** ± 0.037 | **0.9518** ± 0.008 | **0.7615** ± 0.024 | **0.618** ± 0.035 | **0.9097** ± 0.004 | **0.7979** ± 0.006 | **0.6140** ± 0.033 | **0.9661** ± 0.002 |
| Gain | +3.40%* | +11.26%* | +1.14%* | +0.73%* | +0.32%* | +0.84%* | +1.35%* | +1.59%* | +0.33%* |

**Table 8: Hyper-parameter setting.**

| Hyper-parameter | FB15K | TMDB5K | IMDB |
|---|---|---|---|
| $d$ | 256 | 256 | 256 |
| $N$ | 10 | 10 | 10 |
| $H$ | 4 | 4 | 4 |
| $T$ | 5 | 5 | 5 |
| Dropout Ratio | 0.3 | 0.3 | 0.3 |
| learning Rate | 0.005 | 0.005 | 0.005 |
| $L$ | 2 | 1 | 1 |
| $\lambda$ | 1 | 1 | 1 |
| $\mu_1$ | 0.9 | 0.9 | 0.6 |
| $\mu_2$ | 0.1 | 0.1 | 0.4 |

we follow [17] by additionally introducing the listwise learning-to-ranking loss [4] to boost EASING for this task. Specifically, for each importance value label, we first randomly sample $n$ labeled nodes in a set $R(x)$. The learning-to-ranking loss is defined as:

$$\mathcal{L}_{rank} = - \sum_{y \in R(x)} s'_y \log(\widehat{s}'_y), \qquad (33)$$

where $s'_y$ and $\widehat{s}_y'$ are normalized as follows:

$$s'_y = \frac{\text{EXP}(s_y)}{\sum_{j \in R(x)} \text{EXP}(s_j)}, \quad \widehat{s}'_y = \frac{\text{EXP}(\widehat{s}'_y)}{\sum_{j \in R(x)} \text{EXP}(\widehat{s}'_j)}. \qquad (34)$$

We denode the variant as EASING$_{rank}$ and report results in Table 7. We notice that such modification generally leads to positive effects in improving the ranking performance. However, on the other hand, this will also increase the computation overhead [5, 17]. We leave the investigation of lightweight joint training as future work.

## B.2 Hyper-parameter Settings

We report the hyper-parameter settings in Table 8.

## B.3 Hyper-parameter Sensitivity Study

We illustrate experiments on FB15K. As shown in Tables 9-12, we set $T$, $L$, $\lambda$, and $(\mu_1, \mu_2)$ as 5, 2, 1.0, and (0.9, 0.1), respectively. Specifically, $T$ denotes the ensembling times where $T = 5$ can already

achieve satisfactory performance. Setting $L = 2$ also shows a balanced model performance, while a larger value, e.g., $L = 4$, may lead to heavy overfitting problem. As for $\lambda$ and $(\mu_1, \mu_2)$, we simply set it as 1 and (0.9, 0.1), which can effectively derives good performance.

**Table 9: Setting of $T$.**

| $T$ | MAE | RMSE | NRMSE | SPEARMAN | Precision@100 | NDCG@100 |
|---|---|---|---|---|---|---|
| 1 | 0.7406 | 0.9944 | 0.0974 | 0.7459 | 0.4820 | 0.9497 |
| 2 | 0.7396 | 1.0054 | 0.0985 | 0.7459 | 0.4800 | 0.9496 |
| 3 | 0.7327 | 0.9930 | 0.0973 | 0.7535 | 0.5020 | 0.9526 |
| 4 | 0.7359 | 0.9930 | 0.0973 | 0.7466 | 0.4920 | 0.9517 |
| 5 | 0.7315 | 0.9846 | 0.0964 | 0.7519 | 0.4780 | 0.9506 |
| 6 | 0.7426 | 0.9974 | 0.0976 | 0.7430 | 0.5020 | 0.9522 |
| 7 | 0.7317 | 0.9870 | 0.0966 | 0.7521 | 0.5180 | 0.953 |
| 8 | 0.7315 | 0.9874 | 0.0967 | 0.7483 | 0.5020 | 0.9519 |
| 9 | 0.7402 | 1.0018 | 0.0982 | 0.7444 | 0.5020 | 0.9508 |
| 10 | 0.7304 | 0.9815 | 0.0963 | 0.7527 | 0.4740 | 0.9489 |

**Table 10: Setting of $L$.**

| $L$ | MAE | RMSE | NRMSE | SPEARMAN | Precision@100 | NDCG@100 |
|---|---|---|---|---|---|---|
| 1 | 0.7362 | 0.9907 | 0.0970 | 0.7476 | 0.4900 | 0.9507 |
| 2 | 0.7315 | 0.9846 | 0.0964 | 0.7519 | 0.4780 | 0.9506 |
| 3 | 0.7454 | 0.9969 | 0.0976 | 0.7446 | 0.4840 | 0.9519 |
| 4 | 0.7843 | 1.0465 | 0.1035 | 0.7283 | 0.4820 | 0.9493 |

**Table 11: Setting of $\lambda$.**

| $\lambda$ | MAE | RMSE | NRMSE | SPEARMAN | Precision@100 | NDCG@100 |
|---|---|---|---|---|---|---|
| 1.2 | 0.7387 | 0.9935 | 0.0973 | 0.7466 | 0.4900 | 0.9513 |
| 1.0 | 0.7315 | 0.9846 | 0.0964 | 0.7519 | 0.4780 | 0.9506 |
| 0.8 | 0.7354 | 0.9959 | 0.0975 | 0.7449 | 0.4960 | 0.9487 |
| 0.6 | 0.7367 | 0.9963 | 0.0977 | 0.7472 | 0.4840 | 0.9505 |
| 0.4 | 0.7401 | 0.9973 | 0.0978 | 0.7422 | 0.5000 | 0.9507 |
| 0.2 | 0.7361 | 0.9938 | 0.0974 | 0.7423 | 0.4920 | 0.9508 |

**Table 12: Setting of $\mu_1$ and $\mu_2$.**

| $(\mu_1,\mu_2)$ | MAE | RMSE | NRMSE | SPEARMAN | Precision@100 | NDCG@100 |
|---|---|---|---|---|---|---|
| (0.9, 0.1) | 0.7315 | 0.9846 | 0.0964 | 0.7519 | 0.4780 | 0.9506 |
| (0.8, 0.2) | 0.7299 | 0.9866 | 0.0967 | 0.7514 | 0.4900 | 0.9505 |
| (0.7, 0.3) | 0.7313 | 0.9884 | 0.0970 | 0.7501 | 0.4820 | 0.9511 |
| (0.6, 0.4) | 0.7363 | 0.9889 | 0.0970 | 0.7448 | 0.4820 | 0.9502 |
| (0.5, 0.5) | 0.7419 | 1.0018 | 0.0982 | 0.7422 | 0.4900 | 0.9516 |
| (0.4, 0.6) | 0.7354 | 0.9909 | 0.0971 | 0.7459 | 0.4860 | 0.9503 |
| (0.3, 0.7) | 0.7410 | 1.0004 | 0.0979 | 0.7458 | 0.4820 | 0.9510 |
| (0.2, 0.8) | 0.7452 | 1.0007 | 0.0980 | 0.7405 | 0.4880 | 0.9507 |
| (0.1, 0.9) | 0.7391 | 0.9995 | 0.0979 | 0.7440 | 0.4800 | 0.9487 |