# OpenReview forum: "Semi-supervised Node Importance Estimation with Informative Distribution Modeling for Uncertainty Regularization"
_ACM.org/TheWebConf/2025/Conference — WWW 2025 Poster_

### Official Review · Reviewer_j1GP · 2024-11-04

**Novelty:** 5
**Technical Quality:** 4

**Review:**

This paper introduces EASING, the first semi-supervised framework for node importance estimation in heterogeneous graphs, making a significant theoretical and methodological contribution. The DJE architecture uniquely integrates a multidimensional Gaussian distribution for uncertainty modeling with an encoder-decoder design, enabling joint estimation of importance and uncertainty.

Pros:

1.The paper proposes the first semi-supervised framework, EASING, for node importance estimation in heterogeneous graphs, making a pioneering contribution to the integration of semi-supervised learning with graph node importance estimation.

2.The distribution modeling approach and the encoder-decoder structure adopted in DJE are unique in handling node representation and joint estimation of importance and uncertainty.

3.The introduction of uncertainty into the semi-supervised learning process for node importance estimation allows data samples to contribute adaptively to optimization through a heteroscedastic loss term based on uncertainty.

Cons:

1.Some of the experimental settings are not clear, such as the dataset split and other details.

2.The descriptions of the formulas in the methods section are not entirely coherent, with subtle variations in the notation between different parts of the text, which require readers to infer the meaning. Additionally, the encoder uses the multi-head attention mechanism, yet a significant amount of space and formulas are devoted to explaining this existing technique.

3.More details are provided below in the Questions.

**Questions:**

1.Figures 1 and 2 illustrate that not all unlabeled nodes generate pseudo-labels. What is the proportion of nodes with pseudo-labels among all unlabeled nodes? Is there a detailed analysis of this ratio?

2.Based on $H_x$, how are the mean and covariance representations of the distribution derived? Since each node has only one representation vector, how can the mean and standard deviation be obtained from a single vector? Additionally, how are $\alpha_s$ and $\alpha_u$ in Equation 11 computed?

3.Are the $G_x$ and $T_x$ obtained using node2vec and Transformer-XL the same as $G_x^{(0)}$ and $T_x^{(0)}$? After passing through $L$ encoder layers, do they become $G_x^{(L)}$ and $T_x^{(L)}$? Is my understanding correct?

4.Equation 18 mentions the use of two independent DJE structures, while Equation 19 introduces $T$ iterations. What is the difference in focus between "2" in Equation 18 and "$T$" in Equation 19?

5.Could the performance of DJE be pre-trained by leveraging the difference between predictions for labeled nodes and their ground-truth labels? For example, pre-train DJE first, fine-tune it with pseudo-labels, and then continue training. Have you considered showing MSE or similar evaluation metrics for labeled nodes to reflect DJE’s performance?

6.You mentioned that existing studies mainly focus on solving the importance-based ranking problem rather than quantifying exact importance values. However, I noticed that [1] appears to use metrics like MSE, RMSE, and NRMSE for node importance estimation. Would this be considered within the same scope?

7.Among the commonly used datasets in baseline methods, the music10k dataset is not included in the paper. Is there a specific reason for excluding it? Additionally, the node and edge counts in the three datasets used in this paper seem to differ from those reported in the baseline method papers. What is the reason for this?

8.How are the training and testing proportions of nodes determined in the datasets? What does "hidden ground-truth label" mean?

9.Finally, a couple of minor details: there seems to be a potential LaTeX compilation issue in the line following Equation 23, and the legend in Figure 4 obscures some information. Could the legend be moved outside the coordinate box, such as above it?

[1]Deep Structural Knowledge Exploitation and Synergy for Estimating Node Importance Value on Heterogeneous Information Networks

**Reviewer Confidence:**

2: The reviewer is willing to defend the evaluation, but it is likely that the reviewer did not understand parts of the paper

**Scope:**

4: The work is relevant to the Web and to the track, and is of broad interest to the community

---

### Official Review · Reviewer_pgDs · 2024-11-28

**Novelty:** 5
**Technical Quality:** 6

**Review:**

This paper proposes a semi-supervised framework, EASING, for node importance estimation in heterogeneous graphs. The framework introduces a Distribution-based Joint Estimator (DJE) to simultaneously estimate node importance and uncertainty. By generating pseudo-labels for unlabeled data, this method enhances the accuracy of node importance estimation. Experimental results demonstrate significant performance improvements compared to prior methods.

Pros:
- Novel Framework: The paper presents a novel framework leveraging DJE to effectively address node importance estimation in heterogeneous graphs. Unlike existing supervised methods, the proposed framework explicitly captures uncertainty, improving learning quality from partially labeled data. By incorporating pseudo-label generation, the method enriches the training set and enhances estimation accuracy on three real-world datasets.
- Comprehensive Experiments: Comprehensive experiments conducted on three datasets show that the proposed method outperforms state-of-the-art benchmarks, demonstrating its practicality and effectiveness in real-world applications. Substantial empirical evidence shows that EASING achieves significant improvements over existing models in terms of Mean Absolute Error and Root Mean Square Error, with performance gains being statistically significant (p-values below 0.05).
- Ablation Study: The ablation study evaluates the contributions of individual components in the framework, providing insights and guidance for future research. The study shows significant performance drops when the DJE-Encoder and DJE-Decoder components are removed, indicating their critical role in enhancing the model's performance.
- Hyperparameter Analysis: Hyperparameter settings are thoroughly studied and clearly reported, ensuring reproducibility. The detailed hyperparameter sensitivity analysis and comprehensive experimental settings demonstrate a strong commitment to reproducibility, which is beneficial for practitioners.

Cons:
- Computational Overhead: I am not sure whether the proposed method introduces significant computational overhead. While the substantial performance gains mitigate this concern, it is recommended to report the training and inference time for the proposed method and its baseline comparisons to provide a clearer picture of the trade-off between accuracy and computational cost. Including detailed training and inference times for EASING and baseline models would allow for a more complete evaluation of the method's efficiency.
- Static Graph Limitation: The framework is designed for static graphs, and while the potential for extending it to dynamic or temporal graphs is acknowledged, this aspect remains unexplored. Providing a more detailed discussion or future work would be helpful.

**Questions:**

Can the framework be directly applied to dynamic or temporal graphs? If not, what modifications would be necessary to extend its applicability?

**Reviewer Confidence:**

3: The reviewer is confident but not certain that the evaluation is correct

**Scope:**

4: The work is relevant to the Web and to the track, and is of broad interest to the community

---

### Official Review · Reviewer_DZpe · 2024-11-30

**Novelty:** 5
**Technical Quality:** 4

**Review:**

**Summary**

This paper addresses semi-supervised node importance estimation in heterogeneous graphs, focusing on learning with limited labeled data and incorporating uncertainty. The authors propose EASING, a framework with uncertainty regularization, and DJE, an encoder-decoder model for joint importance and uncertainty estimation with pseudo-labeling. Experimental results demonstrate EASING's superior performance over competing methods and the effectiveness of its components on real-world datasets.

**Strong points**
 - S1. Novel semi-supervised framework for node importance estimation.
 - S2. Effective integration of uncertainty modeling with heteroscedastic learning.
 - S3. Demonstrated superior performance on real-world heterogeneous graph datasets.

**Weak Points**
 - W1. The motivation for addressing the problem of importance estimation needs to be further strengthened.
 - W2. The authors need to justify their choice of multidimensional elliptical Gaussian distributions.
 - W3. The experiments require further enhancement.

**Detailed Comments**
 - D1. The motivation for addressing the problem of importance estimation needs to be further strengthened. The authors should explain, from a practical perspective, why importance ranking alone is insufficient to meet real-world demands.
 - D2. The authors need to justify their choice of multidimensional elliptical Gaussian distributions. Additionally, they should clarify whether the proposed approach can adopt other types of distributions, and discuss how replacing the distribution impacts the experimental results.
 - D3. The experiments require further enhancement. Specifically:
   - The TMDB5K dataset has the lowest ratio of labeled training data, yet the performance improvement on this dataset is the smallest. Does this indicate that the semi-supervised importance learning approach failed to show its effectiveness here? More explanations are needed.
   - In Table 5, why does the precision improve in the case of w/o SSIL?
   - The experiments on importance-based ranking should include comparisons with relevant related work, such as [12, 6].
 - D4. In pseudo-label generation, only two DJE are used for ensembling. Why is this the case? Would ensembling more DJE improve the results? Experimental evidence should be provided.

**Questions:**

Please refer to D1-D4.

**Reviewer Confidence:**

3: The reviewer is confident but not certain that the evaluation is correct

**Scope:**

4: The work is relevant to the Web and to the track, and is of broad interest to the community

---

### Official Review · Reviewer_aYxh · 2024-12-02

**Novelty:** 5
**Technical Quality:** 5

**Review:**

The authors propose a framework  EASING to study heterogeneous graph node importance estimation in a
semi-supervised learning setting. They introduce an encoder-decoder system DJE, which constructs
high-quality pseudo labels for initially unlabeled data, thereby enriching the dataset for model training.
The experiments show demonstrate the effectiveness of our proposed methods on three datasets.

**Questions:**

No immediate questions

**Reviewer Confidence:**

1: The reviewer's evaluation is an educated guess

**Scope:**

3: The work is somewhat relevant to the Web and to the track, and is of narrow interest to a sub-community